# OpCode-Level Function Call Graph Based Android Malware Classification Using Deep Learning

**DOI:** 10.3390/s20133645

**Published:** 2020-06-29

**Authors:** Weina Niu, Rong Cao, Xiaosong Zhang, Kangyi Ding, Kaimeng Zhang, Ting Li

**Affiliations:** 1School of Computer Science and Engineering, Institute for Cyber Security, University of Electronic Science and Technology of China (UESTC), Chengdu 611731, China; niuweina1@126.com (W.N.); tutu6289@163.com (R.C.); kangyiding@gmail.com (K.D.); kmengzhang@gmail.com (K.Z.); ryokoside@sina.cn (T.L.); 2Cyberspace Security Research Center, Peng Cheng Laboratory, Shenzhen 518040, China

**Keywords:** Android malware detection, static analysis, OpCode-level FCG, deep learning, LSTM

## Abstract

Due to the openness of an Android system, many Internet of Things (IoT) devices are running the Android system and Android devices have become a common control terminal for IoT devices because of various sensors on them. With the popularity of IoT devices, malware on Android-based IoT devices is also increasing. People’s lives and privacy security are threatened. To reduce such threat, many researchers have proposed new methods to detect Android malware. Currently, most malware detection products on the market are based on malware signatures, which have a fast detection speed and normally a low false alarm rate for known malware families. However, they cannot detect unknown malware and are easily evaded by malware that is confused or packaged. Many new solutions use syntactic features and machine learning techniques to classify Android malware. It has been known that analysis of the Function Call Graph (FCG) can capture behavioral features of malware well. This paper presents a new approach to classifying Android malware based on deep learning and OpCode-level FCG. The FCG is obtained through static analysis of Operation Code (OpCode), and the deep learning model we used is the Long Short-Term Memory (LSTM). We conducted experiments on a dataset with 1796 Android malware samples classified into two categories (obtained from Virusshare and AndroZoo) and 1000 benign Android apps. Our experimental results showed that our proposed approach with an accuracy of 97% outperforms the state-of-the-art methods such as those proposed by Nikola et al. and Hou et al. (IJCAI-18) with the accuracy of 97% and 91%, respectively. The time consumption of our proposed approach is less than the other two methods.

## 1. Introduction

Nowadays, smartphones and other mobile devices are used on a widespread in our daily life. International Data Corporation (IDC) expected 77.7% of networked devices to be smartphones by 2019 [1], and the market share of Android smartphones was 70.69% in April 2020 [2]. Resulting from the prevalence of Android devices and the openness of the Android OS, many Internet of Things (IoT) devices are running the Android system. Moreover, mobile devices based on an Android system have become a commonly used control terminal for IoT devices due to their various sensors. However, the amount of Android malware has been rapidly increasing. By the end of 2018, Kaspersky’s Mobile malware evolution 2019 [3] showed that about 9599 malicious installation packages appeared per day. Mobile devices infected with malware can cause leakage of important private information such as users’ accounts and passwords. Moreover, the rise of tariff-consuming and money-stealing malware can lead to economic losses of users. The Android malware special report from 360 Internet Security Center [4] in 2020 indicated that the monetary loss per victim caused by fraud in China is as high as 8218 RMB (over 1149 USD, based on the exchange rate on 28 May 2020). Furthermore, for Vehicle Internet of Things, if the system is controlled by attackers, people’s lives and privacy will be threatened. Therefore, it is quite urgent to effectively detect and defend Android malware.

Current mainstream Android malware detection approaches identify and classify malicious apps through machine learning techniques taking different features (mainly API call features, assembly code features, and binary code features) as the input. Similar to general malware detection methods, current Android malware detection methods can be divided into two main categories, namely, signature-based and behavior-based methods. The first type of methods classifies malicious Android apps according to some unique digital signatures of known malware types. Such signatures are generated by performing static analysis of known malware samples to review their code structures, which include features at the assembly code and binary code levels. However, signature-based methods normally cannot detect unknown Android malware or even more complicated malware such as polymorphic malware. The second type of methods executes a given Android malware sample in a sandbox and get its runtime behavioral information from which relevant Application Programming Interface (API) calls features and assembly code features can be extracted. Such behavior-based approach is called dynamic analysis. However, dynamic analysis based methods naturally have a higher overhead in running time, and it can miss some hidden malicious behavior (if not triggered during the monitoring period). The performance of Android malware detection methods based on machine learning depends on how well the extracted features represent differences between different types of Android malicious apps and benign Android apps. Features used by existing Android malware detection methods are often too simple, thus limiting the accuracy of detection.

In this work, we propose to use deep learning with static analysis of the OpCode-Level Function Call Graph (FCG) to explore more complicated features of Android malware. Deep learning is a new frontier of machine learning and data mining with superior ability in feature learning through multilayer deep architecture, and deep learning can evolve high-level representations by associating the static and dynamic features in Android malware. Some scholars [5,6,7] have worked with the deep learning methods in Android malware detection and all achieve an accuracy rate of more than 90%, which show that deep learning is especially suitable for characterizing Android malware. Specifically, we use the FlowDroid [8] tool to generate the FCG for a given Android app sample (a .apk file), and get the class files of different functions, then we reorganize the Dalvik bytecode according to the calling order of the functions. Finally, we use the Long Short-Term Memory (LSTM) model [9] to classify the given Android sample. The main contributions of this paper can be summarized as follows:We propose to combine deep learning and FCG graph analysis for Android malware classification.We use the OpCode-level FCG which can better capture the intrinsic association between code and get higher semantic link information to fine-grain characterize behaviors of Android programs and construct the Android malware discriminant model based on LSTM, which fully considers the relevance and temporal of malicious program behaviors.We also design Android malware classification rules based upon the recognition results of AVTEST [10] and VirusTotal [11] during construction of the experimental dataset.

The remainder of this paper is arranged as follows: Section 2 discusses the related work, followed by several important concepts behind our proposed approach that are described in Section 3. In Section 4, our proposed Android malware detection approach is demonstrated in detail, followed by experimental results analysis of our proposed approach being presented in Section 5. Some conclusions are summarized in Section 6.

## 2. Related Work

Automatic Android malware detection has become an issue in research now. A lot of research [12,13,14] has been done and many ways have been brought forward to detect malware based on machine learning. Their detection process has two steps: feature extraction and classification. Three different types of methods have been frequently applied in the step of feature extraction: static analysis, dynamic analysis, and hybrid analysis. We review some selected related work in each of the three types in this section.

### 2.1. Static Analysis

Static analysis based methods analyze the software source code or binary code from the perspectives of syntax and semantics without running the software. Such methods have the general advantages of faster detection speed and lower false positive rate. However, a common shortcoming of static analysis methods is that they cannot effectively detect malware that uses obfuscation techniques such as packing, encryption, compression, and deformation.

Sanz et al. [15] classified Android applications into several classes according to features extracted from Android App Market and application itself. App-specific features consist of requested permissions and strings contained in the application. Market-specific features contain app ratings, number of ratings, and the size of application. After extracting these features through the Android Market API, they verified the accuracy of their method using several supervised machine learning techniques. Their experimental results revealed that the Area Under Curve (AUC) of Tree-Augmented Naive (TAN) Bayesian classifier was 0.93. Notwithstanding, the sample size for their experiment was only 820.

Wu et al. [13] developed a static analysis based system, DroidMat, to detect Android malware. They treated static information (such as permissions, component deployment, Intent messaging, and API calls) as the basis for characterizing the behaviors of Android apps. DroidMat extracted information such as the requested permissions from each app’s manifest file and treated it as an entry point for tracking API calls associated with permissions. Then, it identified Android malware using a k-Nearest Neighbor (k-NN) algorithm. Unfortunately, DroidMat couldn’t perform well on detecting two Android malware families: BaseBridge and DroidKungFu.

Daniel et al. [16] came up with a system called Drebin to detect malware using lightweight static analysis. Drebin statically analyzed a given Android app and collected a number of features such as API calls and requested permissions. Then, it used a Support Vector Machine (SVM) to determine if the analyzed app is malicious. Drebin can not only effectively scan a lot of applications, including apps from third-party application markets, but also scan directly on mobile devices. Their experimental results showed that Drebin had achieved an average accuracy of 93% and a very low FPR (false positive rate) of 1%. Nevertheless, renaming components and activities between training and detection phases may influence discriminant features’ effectiveness. An attacker could also reduce the detection accuracy of Drebin through integrating fake invariants or benign features into malware.

Hou et al. [17] developed DroidDelver, which was a system which can automatically detect Android malware. DroidDelver extracted features of an API call block from smali code and used a Deep Belief Network (DBN) to detect Android malware. Their results demonstrated that the recognition rate of DroidDelver achieved 96.66%. DroidDelver can be further improved by increasing the semantic depth of feature extraction.

Milosevic et al. [18] proposed two methods for using machine learning to aid static Android malware analysis. The first method focuses on permissions, and the second method makes use of source code analysis combined with a Bag-of-Words (BoW) model. The experimental results of these two methods revealed that their F-values were 89% and 95.1%, respectively. However, both methods require a lot of computational resources in the analysis process.

MalDozer [19] was an app API calls based multi-class Android malware classification framework. The framework examined behaviors of a given app from the sequence and patterns of API calls. MalDozer first extracted the API call sequence from the DEX file, then classified and discretized the sensitive APIs, and finally applied a Deep Neural Network (DNN). Through different data sets and the different number of cross-validation folds, the lowest F1 reached 95.0252%, and the highest reached 99.8482%. MalDozer was proved to be highly accurate in malware detection including attribution to the correct malware families. MalDozer is less affected by obfuscation techniques, but it is not immune to dynamic code loading and reflection confusion methods.

### 2.2. Dynamic Analysis

The dynamic analysis process needs to run applications in a closed environment and analyze the behavioral characteristics of the application. Dynamic analysis is more complicated than static analysis since it takes a long time to acquire the dynamic data of the application that runs in real time.

DroidDolphin [20] utilized some dynamic analysis tools, including DroidBox [21] and APE (A Smart Automatic Testing Environment for Android Malware) [22], to extract 13 activity features from each Android app. Then, they apply SVM to establish the prediction model. The results of the test indicated the accuracy of DroidDolphin had reached 86.1%, and its F value (a measure of combined accuracy and recall) was 85.7%. However, its detection efficiency was low and the detection time consumed was long. In addition, it could not detect malware with anti-emulation technology.

CopperDroid [23] is a dynamic analysis system that adopted the automatic Virtual Machine Introspection (VMI) technique. It extracted interactions between the app and the operating system (e.g., requests to create files and processes), as well as inter-process and in-process communications such as SMS receptions as features for behavior representation of the target Android app. CopperDroid successfully disclosed more than 60% malicious behaviors of 1365 malware samples. The method has a limitation that it can only automatically check parameters of methods contained in the interface that owns the AIDL file. As a general input generator, IntelliDroid [24] was used for triggering malicious behaviors of malware Android apps to facilitate dynamic analysis. There are many Android portals composed of many event handlers in an Android app. It is likely that multiple handlers can be triggered in a certain order to wake up some malicious behavior. Experimental results indicated that the average execution time of the tool was 138.4 s, and 70 of 75 malware behaviors could be triggered by its generated input. Currently, it can detect only data dependencies among events. If malicious behavior relevant to heap variables occurs, IntelliDroid cannot generate input for data streams by files.

### 2.3. Hybrid Analysis

There are also some reported methods that detect Android malware through a combination of dynamic analysis and static analysis.

StaDynA [25] interacted with the static analysis and dynamic analysis to scrutinize Android app with reflection confusion and dynamic class loading. This method extended the app’s Method Call Graph (MCG) depending on other runtime loading modules and other execution paths hidden in reflection invocations. StaDynA can generate extended call graphs without modifying the app. The results of StaDynA were imported into the latest analyzers to raise their recognition rate. For example, reachability analysis of the scalable MCG would be more accurate than the original MCG. However, StaDynA cannot effectively handle multiple apps using the same User Identification (UID).

DroidDetector [26] was a detection engine that combines static analysis with dynamic analysis and uses Deep Belief Networks (DBN) architecture to characterize Android malware. After a .apk file was submitted, DroidDetector checked its integrity and determines if it is complete, correct, and legitimate. Next, DroidDetector performed static analysis to obtain permissions and sensitive API calls requested by this app. DroidDetector then performed dynamic analysis by installing and running the app in the Droid-Box [21] for a fixed amount of time. Finally, DroidDetector combined static and dynamic features to classify the given Android app. Experimental results showed that it had achieved a detection accuracy of 96.76%. However, its feature extraction granularity is too coarse and the model is not comprehensive enough. For example, we can use a richer discrete feature rather than a binary feature to build a deep learning model.

Hou et al. [27] not only used API calls but also analyzed the different relationships between them. Android applications themselves and their related APIs, as well as their rich relationships, were expressed as a structured heterogeneous information network (HIN). They made use of a method based on Multi-Kernel learning to classify the Android apps. Experimental results showed that they achieved an accuracy of 98.8%. However, their HIN is based on the API level, which is higher than OpCode-Level FCG.

## 3. Preliminaries

Long Short-Term Memory (LSTM), as a special type of Recurrent neural network (RNN) [28], is able to learn long-term dependency information. In a standard RNN, duplicate modules have only a very simple structure, including a linear operational and an activation function. LSTM follows the basic structure of an RNN, but duplicate modules have well-designed structures. LSTM includes four layers with interactive processing and delivery information. LSTM is a well-designed network structure with three “gate” structures, which is shown in Figure 1.

The first step of LSTM is deciding what information should be discarded from the cell state. This step is implemented through Forgotten Gate which is an activation function called sigmoid. The forgotten gate reads hidden information from the last moment $h_{t-1}$ and current input information $x_{t}$, and calculates a value between 0 and 1, this number represents the degree of $C_{t-1}$ lost. 1 means “completely reserved” and 0 means “completely discarded”. The forgetting gate expression is:(1)$$f_{t}=\sigma \left(W_{f}.\left[h_{\left(t-1\right)},x_{t}\right]+b_{f}\right)$$

Here, $h_{t-1}$ indicates output hidden information about the previous state, $x_{t}$ indicates input information of the current state, and σ represents activation function called sigmoid.

The second step of LSTM is deciding how much current information to add to $C_{t-1}$. This step is implemented by the Input Gate which has two parts: an activation function called sigmoid determines what information requires to be added; a tanh layer generates a vector $\tilde{C_{t}}$, which is the alternative content for updating. Finally, these two parts are combined and each of the cells is updated as follows:(2)$$i_{t}=\sigma \left(W_{i}.\left[h_{\left(t-1\right)},x_{t}\right]+b_{i}\right)\;\tilde{C_{t}}=\tanh\left(W_{C}\right).\left[h_{t-1},x_{t}\right]+b_{C})$$

Then, LSTM updates the old cell state $C_{t-1}$ to new cell state $C_{t}$ as follows:(3)$$C_{t}=f_{t}*C_{\left(t-1\right)}+i_{t}*\tilde{C_{t}}$$

The third step of LSTM is determining the output value based on the cell state. The output gate runs a sigmoid function to determine which part of the cell state will be the output. Next, the output gate processes the cell state through the tanh function (with a value between –1 and 1 as the input) and multiplies it by the output of the sigmoid function. Eventually, it will only output the part that is determined. The equations are as follows:(4)$$o_{t}=\sigma \left(W_{o}\left[h_{\left(t-1\right)},x_{t}\right]+b_{o}\right)\;h_{t}=o_{t}*\tanh\left(C_{t}\right)$$

## 4. Proposed Detection Method

This section illustrates the proposed Android malware detection method whose goal is to detect Android malware and classify any detected malware sample into a malware category. The framework of our proposed detection method in this work is shown in Figure 2, which has two phases: training (i.e., learning) and detection. The input of the training phase is a large number of training Android apps, some of which are malware and the others are not. The output of the training phase is a trained LSTM-based neural network.

### 4.1. The Training Phase

As highlighted in Figure 2a, the training phase has five steps: constructing dataset, pre-processing, generating OpCode sequences based on FCG, transforming into vector and training LSTM-based neural networks.

#### 4.1.1. Constructing Dataset

In the dataset construction step, we mainly labeled Android malicious samples. According to Tencent Mobile Security Lab 2018 Mobile Security Report [29], two categories of Android malware (“Adware” and “Trojan”) appeared more frequently in the real world. Thus, we decided to divide all Android malware samples into three categories: “Trojan”, “Adware”, and “Others”. However, we did not use the “Others” category for training because the “Others” category contains too many families, and the number of each family is not balanced enough to effectively learn the corresponding features. Android malware classification rules based on the results of AVTEST, VirusTotal, and Euphony [30] have four sub-steps.

Sub-step 1: Using the VirusTotal website that integrates 88 virus detection engines to check all samples for binary results (malware or not).Sub-step 2: For malicious samples, since each virus detection engine has different naming rules for malicious families, we use the report obtained from Virustotal as the input of the tool Euphony, and obtained a standardized intermediate result (each detection engine determines what type the sample is).Sub-step 3: The detection engines on VirusTotal are scored according to the AVTEST website’s evaluation results in January 2019. The higher the engine detection effect, the greater the weight. Thus, we set the weights of prominent engines to an arithmetic progression. Weights of prominent detection engines are shown in Table 1. The weight of the remaining engines is 0.Sub-step 4: Choosing the largest result of the final score and mapping it to our classification category. The formula is expressed as:
(5)$$family=\{y|max\left(score\left(y_{1}\right),score\left(y_{2}\right),score\left(y_{3}\right)\right)\}$$
where
(6)$$score\left(y_{i}\right)=weight\left(x_{1}\right)+weight\left(x_{2}\right)+...+weight\left(x_{88}\right)$$
where $y_{i}$ represents the $i_{th}$ malware category, $x_{j}$ represents the $j_{th}$ engine on VirusTotal, $weight(x_{j})$ indicates the weight of the $j_{th}$ detection engine whose detection result is $y_{i}$.

#### 4.1.2. Pre-Processing

In the pre-processing phase, we first unpacked the Android apps by generic unpackers like drizzleDumper and FUPK3. Then, we performed the other two main processes:Process 1: Transforming Android apks into OpCode representations. This process has two sub-steps, which is highlighted below.-Sub-step 1: Using unzip to decompress .apk files to obtain their .dex files. This step aims to get the classes.dex file by decompressing the Android .apk file.-Sub-step 2: This step aims to get Dalvik code from the corresponding APK using disassemble tools (e.g., dedexer [31]). This is necessary in order to use OpCode-level features.Process 2: Generating the FCG of different training programs through the FlowDroid tool, which automatically extracts API calls from decompiled smali codes and converts to static execution sequences of corresponding API calls. This process is highlighted in Figure 3. The FCG is denoted by G=(V,E), where $V=\{v_{i}|1\leq i\leq n\}$ represents the set of functions called by Android malware sample, E⊆V∗V is the set of function call relationships, and the edge $\{v_{i},v_{j}\}\in E$ indicates that there is a function call relationship from the caller $v_{i}$ to the caller $v_{j}$.

#### 4.1.3. Generating OpCode Sequences Based on FCG

Based on the extracted API calls, the OpCode-level FCG can be obtained. Starting with the first API, locating the class file it belongs to, and then finding the corresponding method by traversing the contents of the .ddx files in the class, and finally extracting the OpCode sequence between the method and the end method. Figure 4 shows an example of the generating OpCode sequences based on FCG step.

#### 4.1.4. Transforming into a Vector

This step is used to encode the symbolic representations of apps into vectors. Using one-hot code which is a process by which categorical variables are converted into a form that could be provided to ML algorithms to do a better job in prediction to transform generated OpCode sequences based on FCG into the vector.

#### 4.1.5. Training an LSTM-Based Neural Network

Program language and natural language have some similarities, such as both having grammatical information and semantic information [32]. Thus, it can be processed using Natural Language Processing (NLP) tools. We use deep learning instead of machine learning, although machine learning is more suitable for data sets with small sample size because, for simple tasks, such as the three classifications of Android software we do, the appropriate deep learning model can also achieve good results with fewer samples. Moreover, the opcode sequence obtained from Android software after being decompiled is sequential, while RNN is designed to process the data with temporal characteristics. LSTM is improved on the basis of RNN. The unique threshold mechanism can alleviate gradient disappearance and gradient explosion, and obtain long-term dependent information through the cell state. Thus, we establish the LSTM-based recurrent neural network and use the grid search and cross-validation to adjust the parameters. The network structure of our trained LSTM model is shown in Figure 5, whose parameters of the embedding layer are initialized through Word2Vector. The Word2Vector includes four sub-steps:Sub-step 1: Statistics and processing of opcode sequences. Counting the vocabulary and using the lexical word frequency to number the vocabulary, adding the default vocabulary and dealing with the special vocabulary, etc..Sub-step 2: Building (data, tag) pairs. Constructing *skip_gram* training data. *skip_gram* is a kind of data constructor, which utilizes the center word for predicting their surrounding words.Sub-step 3: Establishing the Word2Vec neural network, and using *nce_loss* as the loss function to train. When the training is completed, the parameters of the network are an embedded matrix.Sub-step 4: Transforming into a word embedding format through the obtained embedded matrix.

### 4.2. The Detection Phase

As highlighted in Figure 2b, given one or multiple target apps, we unpack the Android apps by generic unpackers like drizzleDumper and FUPK3 and extract API function calls from them as well as the corresponding OpCode, which is assembled into OpCode sequences based on FCG. The OpCode sequences based on FCG are transformed into their symbolic representations, which is encoded into vectors and used as an input of our trained LSTM-based neural network. The network output vectors are “Trojan” (“1”), “Adware” (“2”) or “Normal” (“0”). This phase has two steps:Step 1: Transforming target apps into OpCode sequences based on FCG and vectors. It has four sub-steps:-Sub-step 1: Obtaining the Dalvik code from the target apps (similar to 4.1.2. Process 1 ).-Sub-step 2: Extracting API function calls from the target apps (similar to 4.1.2. Process 2).-Sub-step 3: Assembling the program slices into OpCode sequences based on FCG (similar to 4.1.3).-Sub-step 4: Transforming OpCode sequences based on FCG into vectors (similar to 4.1.4).Step 2: Detection. This step uses the learned LSTM-based neural network to classify the vectors corresponding to the OpCode sequences based on FCG that are extracted from the target apps. When a vector is classified as “1” (i.e., “Trojan”), it means that the corresponding app is malware and the category is “Trojan”. When a vector is classified as “2” (i.e., “Adware”), it means that the corresponding app is malware and the category is “Adware”. Otherwise, the corresponding OpCode sequences based on FCG is classified as “0” (i.e., “Normal”).

## 5. Experimental Results Analysis

In this section, we introduce the experimental dataset, experimental settings, as well as performance metrics used to verify performance of our proposed detection method. We then discuss experimental results in detail.

### 5.1. Dataset

To evaluate our method, we used real-world Android apps including benign and malicious apps in .apk format. We collected 1000 benign .apk files from Android app store named Pea Pods for major mobile phone manufacturers. We scanned all the .apk files using VirusTotal to ensure that they have no malicious behaviors. We used 1796 .apk files with malicious behaviors, collected from VirusShare [33] and AndroZoo [34] where millions of Android apps collected and arranged by research communities. We collected these samples which included variants of the same sample at different phases from different platforms and at different times. Evaluating our method on these samples will make our results more general and convinced.

As mentioned in Section 4.1.1, we use two Android malware categories: “Trojan” and “Adware”, of which the numbers are 1000 and 796, respectively.

In the end, we randomly shuffled the data set, and then set the ratio of the number of training set to test set to 3 to 1. The training set was used in the training phase, and finally tested the effectiveness of our method on the test set.

### 5.2. Experimental Settings

We explain experimental settings for evaluating the effectiveness of our proposed Android malware detection method. The experiment was conducted software written in or working with Python 3.6, Jupyter 1.0.0, NumPy 1.14.5, Keras 2.2.4, scikit-learn 0.19.1 [35], TensorFlow 1.11.0 [36]. We compared different classifiers (including the SVM-based and LSTM-based model) in the experiments.

The SVM-based classifier is highlighted in Figure 6 and elaborated below, which has two steps different from the LSTM-based classifier in the training process:Step 1: Transforming OpCode sequence based on FCG of the app into vector representations, which has three sub-steps.-Sub-step 1: Generating feature dictionary. Counting the OpCode that has appeared in the training dataset, and using the OpCode set as a feature set.-Sub-step 2: Getting the vector representations of different apps. Counting times of different OpCode appears in different apps and using them as the corresponding feature values.-Sub-step 3: Normalizing frequency features. This is essential to prevent the impact of enormous frequency feature differences on model training due to differences in program size.Step 2: Establishing the SVM-based classifier and adopting the grid search and cross-validation to adjust parameters.

In order to evaluate the true positive rates and false positive rates, we use the GridSearch technology in scikit-learn to automatically adjust the parameters. First, the neural network model is defined as a function, and then the class KerasClassifier is used to wrap the model, and then the function GridSearchCV is used to implement network search. By specifying the attribute cv to be 10 and using 10-fold cross-validation, the neural network model runs 10 epochs and stops when the loss no longer drops. Finally, the optimal parameters selected are shown in Table 2 and Table 3.

There are two very important parameters, *max_path* and *max_sequence*, where *max_path* indicates the maximum number of times each program runs, and *max_sequence* indicates the maximum number of opcodes selected in each run path. Because the number of paths in a program and the number of opcodes for a path is inconsistent, we need to set the threshold. Figure 7 illustrates detection performances for different thresholds of max-path and max-sequence.

As shown in Figure 7a, when max_path increases from 50 to 150, F1-score increases, it is deduced that scores of max_path are greater than 150. This is explained by model overfitting. On the other hand, Figure 7b compares the effects of difference max_sequence. Overall, when the value of max_sequence is greater than 600, F1-Score decreases. Thus, the value of max_path is set to 150 and the value of max_sequence is set to 600 in our experimental environment.

### 5.3. Evaluation Metrics

We used widely used metrics to evaluate the performance of our detection method, which is defined as follows:(7)$$\begin{array}{cc} \; & Precision=TP/(TP+FP)\\ \; & Recall=TP/(TP+FN)\\ \; & TPR=TP/(TP+FN)\\ \; & FPR=FP/(TN+FP)\\ \; & F1-Score(F-measure)=2*Precision*Recall/(Precision+Recall) \end{array}$$
where TP (True Positive) is the number of malicious Android apps that are correctly labeled as malicious, FN (False Negative) is the number of benign Android apps that are falsely labeled as malicious apps. FP (False Positive) is the number of malicious Android apps that are incorrectly labeled as benign. The ROC [37] (Receiver Operating Characteristic) curve is a curve in which the horizontal axis is the FPR (false positive rate) and the vertical axis is TPR (true positive rate). The AUC value is the area covered by the ROC curve. The larger the AUC, the better the classification effect. In general, the higher the Precision, Recall, and AUC, the better the recognition effect. Because F1 and AUC combine the results of Precision and Recall, they can comprehensively explain the overall performance of the model, the higher F1 and AUC may explain why the identification approach is more effective. In the final experimental analysis, we focus on these two evaluation methods: F1 and AUC.

### 5.4. Results

Android malware and its variants in the same family follow a similar pattern on sensitive API calls, so we can also focus on the representation of Android malware behavioral features on sensitive API calls. To evaluate the proposed method, we compare different classification models combined different features from the API name, the sensitive API name, the OpCode sequence of FCG, and the opcode sequence of sensitive FCG whose API is sensitive API. In 2014, Rasthofer et al. [38] proposed a novel machine learning guided approach named SUSI to identify sources and sinks directly from the code of any Android API call. We obtain 11,898 Android sensitive functions from the links given in Rasthofer et al.’s paper. Based on the list of sensitive API calls we obtained, we convert the FCG to a sensitive API call graph (i.e., sensitive FCG): $G^{\prime }=\left(V^{\prime },E^{\prime }\right)$, whose conversion rules are as follows:(8)$$\begin{array}{cc} \; & V_{g}=\left\{v_{i}|\exists v_{i}\in V_{s},0<dis\left(v_{j},v_{i}\right)<n,v_{j}\in V\right\}\\ \; & V^{\prime }=V_{s}\cup V_{g},E^{\prime }=\left(V^{\prime }*V^{\prime }\right)\cup E \end{array}$$
where $dis(v_{j},v_{i})$ returns the length of the shortest path from node $v_{j}$ to node $v_{i}$ and $V_{s}\subset V$ is the set of sensitive API calls invoked by the app.

The illustration of the generated features from the API name, the sensitive API name, and the OpCode sequence of sensitive FCG are shown in Figure 8.

In order to more clearly represent these features and the corresponding classification methods used, we will use the following rules for abbreviations, which will be used hereinafter to represent each method:Name of ordinary function call:Bag-of-Words + svm: NOFC-BoW&SVM.Name of sensitive function call:Bag-of-Words + SVM: NSFC-BoW&SVM.Name of ordinary function call:Deep learning model-LSTM: NOFC-LSTM.Name of sensitive function call:Deep learning model-LSTM: NSFC-LSTM.Opcode sequence of ordinary function order:Bag-of-Words + SVM: OOFO-BoW&SVM.Opcode sequence of sensitive function order:Bag-of-Words + SVM: OSFO-BoW&SVM.Opcode sequence of ordinary function order:Deep learning model-LSTM: OOFO-LSTM.Opcode sequence of sensitive function order:Deep learning model-LSTM: OSFO-LSTM.

The experimental results are shown in Figure 9 and Table 4, and we draw the following conclusions:The detection accuracy of the malware recognition method NSFC-LSTM has no significant improvement compared to NOFC-LSTM. However, the detection accuracy of the detection method OOFO-LSTM is superior to OSFO-LSTM. Selecting only sensitive features destroys sequence information because a malicious behavior is usually formed by a series of common operations and sensitive operations. Even sensitive operations are built on normal operations, LSTM cannot just build a perfect sequence model through sensitive operations, resulting in poor results. Thus, LSTM cannot just build a perfect sequence model through sensitive operations.The detection accuracy of the malware recognition method NSFC-BoW&SVM has no significant improvement compared to NOFC-BoW&SVM. The detection accuracy of the detection method OOFO-BoW&SVM is also superior to OSFO-BoW&SVM.Using the sensitive functions to prune the path does not optimize the classification effect. Conversely, features from opcode sequences based on function order have a better classification result than that using path pruning because pruning path often destroys sequence information of malicious behavior, which is usually formed by a series of common operations and sensitive operations.For features from the opcode order based on the function call, the LSTM model has better performance than the BoW&SVM model. For features from the function call path text, the BoW&SVM model achieves a better experimental result than LSTM model. For opcode features: The opcode features have a long chain of dependencies, and the LSTM is designed to solve long-dependency problems. However, the number of Android opcodes in the BoW model is too small, resulting in too small feature dimensions and insufficient representation capabilities. For function name features: When LSTM processes each path, most of the path sequence length is short, which can not play the advantage of LSTM in long sequence processing, and the BoW model is better at this text type feature.Combining graph analysis (function call graph) with program analysis (Dalvik opcode) can achieve better experimental results than graph analysis alone or program analysis alone. The function call graph is used to structure the program, which has strong anti-killing instruction level confusion resistance. Using the Dalvik opcode can describe the program behavior mode in a fine-grained manner. Combining the advantages of both can significantly improve the accuracy of the general behavior of the Android malware family, and thus improve the efficiency of recognizing and classifying large-scale Android malware.

We conducted some experiments to evaluate the performance of our proposed method for detecting Android malware. We also compared with two other methods proposed in recent years using the same dataset described above.

Competing Method 1 [18]: In 2017, Milosevic et al. presented two methods that based on app permissions and source code analysis with the help of machine learning (classification and clustering) to detect and analyze malicious Android apps.

Their method used the word bag model to extract source text feature and utilized the bagging technology to combine multiple machine learning algorithms for improving the detection accuracy. The experimental results are shown in Table 5.

The data in Table 5 show that detection metrics all have good performance, the highest Precision and F1-Score is 0.98, and the Recall and AUC value both reaches 0.99. However, the precision of Class 0 is not good enough. The main reason is that benign Android applications have different types, but the other class is one of certain malware types. In addition, the precision of Class 0 is not good enough.

Competing Method 2 [27]: In 2018, Hou et al. not only used API calls, but also analyzed the different relationships between them. They used structured heterogeneous information network (HIN) to represent the Android applications, related APIs, and their rich relationships. Then, they used a method based on Multi-Kernel learning to classify the Android apps and they claimed they achieved an accuracy of 98.8%. In the process of recurring, since the paper does not introduce the implementation details, we reproduce it in combination with the paper and our own understanding. When we extract API calls, we find that there are hundreds of thousands of API calls in just one hundred apks. However, they extracted only 200 API calls and didn’t introduce how to reduce the number of API calls. Thus, we pick the 2000 API calls that have the highest frequency. We also use the shogun [39] tool to implement multi-kernel learning. Since the value of the obtained meta-path element is too large, each meta-path data are normalized. We set the p-norm parameter as 1 and use the Gaussian kernel. The experimental results are listed in Table 6, from which we can see that the Precision is 0.91, the Recall, and F1-Score both are 0.92, the AUC value is 0.94. However, we find that benign software is less accurate. The main reason is the same as competing Method 1.

Next, our proposed method is compared with the comparative experiments. We use the opcode sequence of ordinary function order as features and adopted LSTM to classify applications, which has the best detection results from Figure 9. The experimental results of our proposed method and the two competing methods are shown in Figure 10 and Table 7. The detection accuracy of our proposed method is largely the same as Milosevic et al.’s method. However, Milosevic et al.’s method uses the source code word feature, so it consumes more time than our proposed method. When applied to situations such as offline detection as an app and scanning one million apks, Milosevic et al.’s method consumes nearly half an hour more than our method. Furthermore, the size of the model Milosevic et al.’s method used is larger than ours so the app size is larger, which causes the user to use more memory to install the app, and the user experience is not good enough. Meanwhile, considering the experiment adopts the bagging method, our proposed method only uses one SVM model; if it does not consider the influence of the bagging method, our method is better than the comparison experiment 1. Compared to Hou et al.’s method, our method has better performance because they just extract 200 API calls. However, when we recur the experiment, we extract 2000 API calls. The number of API calls is not large enough to represent the apks’ behavior and the action to reduce the number of API calls can also cause the loss of some important features. In addition, Hou et al.’s method consumes much more time and their model size is much larger than our method.

## 6. Conclusions

Currently, many researchers are involved in how to curb the rapid growth of Android malware, but most of the methods they proposed were based on malware signatures, which can not effectively detect malware which is unknown, confused, or packaged. The Android malware detection method proposed in this work analyzes the calling sequence of the function call graph between classes and functions in the execution process. It then generates the Dalvik opcode graph according to the function call sequence, which describes the general behavior of Android malware in fine granularity. Finally, we use the deep learning model LSTM to classify Android apps. Our proposed method avoids the difficulty of acquiring behavioral feature for static analysis caused by obfuscation technologies such as malware distortion and polymorphism. It also avoids the space resource occupation caused by dynamic execution of malware analysis and the possible impact of the malware execution system. The experimental results showed that the proposed method has better performance than Naser et al.’s and Hou et al.’s methods and a comparable performance as Milosevic et al.’s method. Our work indicated that the combination of program analysis and graph analysis can achieve a better performance than using either alone. Although the function-based opcode sequence generally works well with .apk files, there are some problems such as function nesting. In the future, we will consider the problem of function nesting and investigate more fine-grained function-based opcode sequence to further optimize the performance of the proposed approach. Furthermore, we will try to put forward a solution for the problem of insufficient data sets caused by sample imbalance. Thereby, adding more sample families and detecting more malware of different families. In addition, we will try to solve the problem of the “Others” category mentioned in Section 4.1.1. Finally, for the risk that the adversarial AI attack will affect our system’s result, we will combine multiple methods, such as using input gradient regularization to improve the robustness of our system [40] and optimizing the deep learning model from multiple angles to defend against this attack.

## Figures and Tables

**Figure 1 sensors-20-03645-f001:**
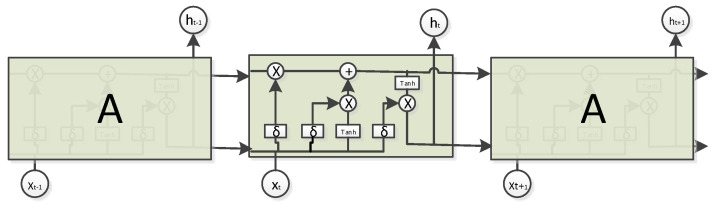
The structure of LSTM.

**Figure 2 sensors-20-03645-f002:**
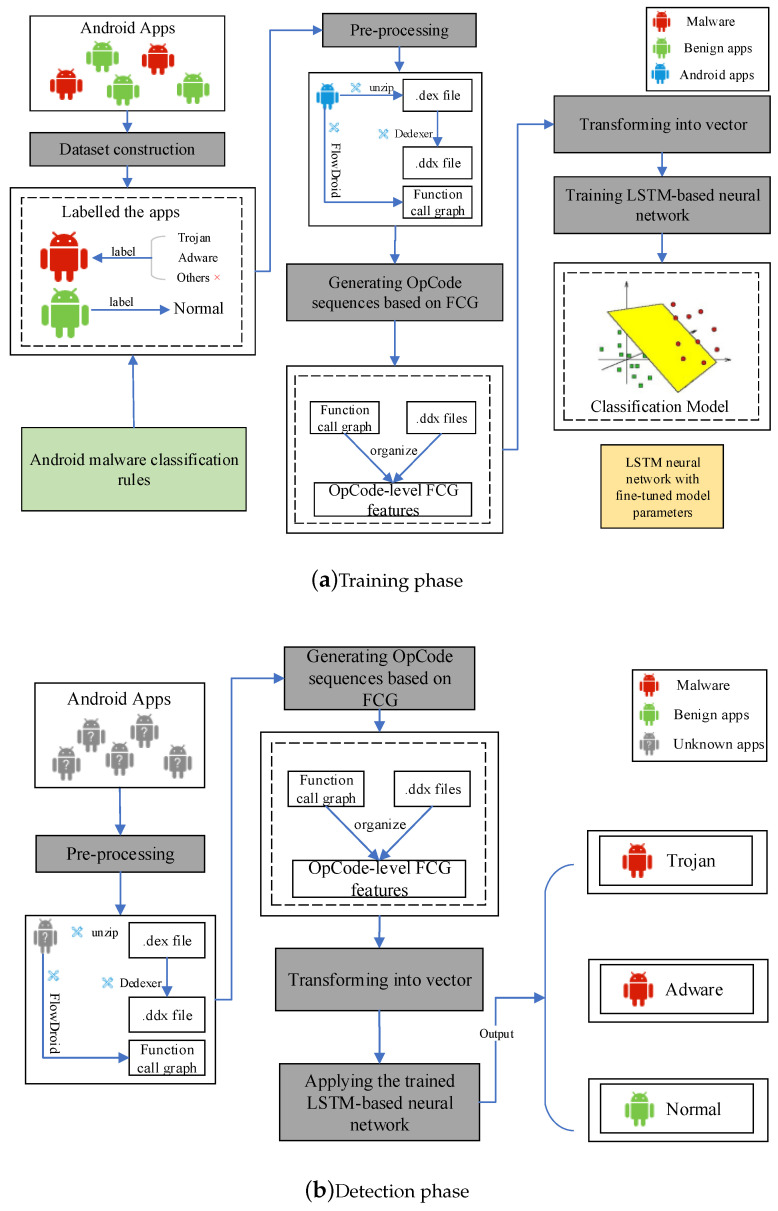
The framework of our proposed Android malware detection method. (**a**) Training phase; (**b**) Detection phase.

**Figure 3 sensors-20-03645-f003:**
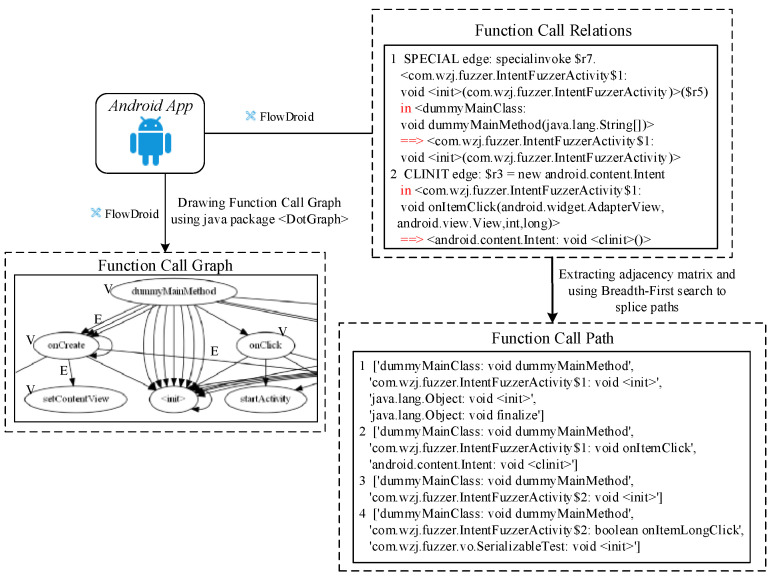
Overview of the generation of the FCG process.

**Figure 4 sensors-20-03645-f004:**
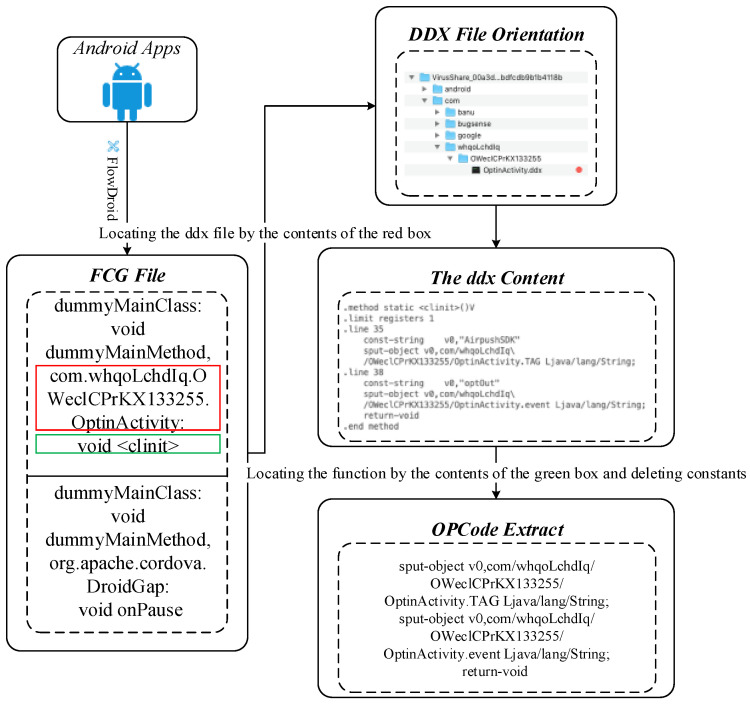
Illustrating the generating OpCode sequences based on FCG from a (training) app.

**Figure 5 sensors-20-03645-f005:**
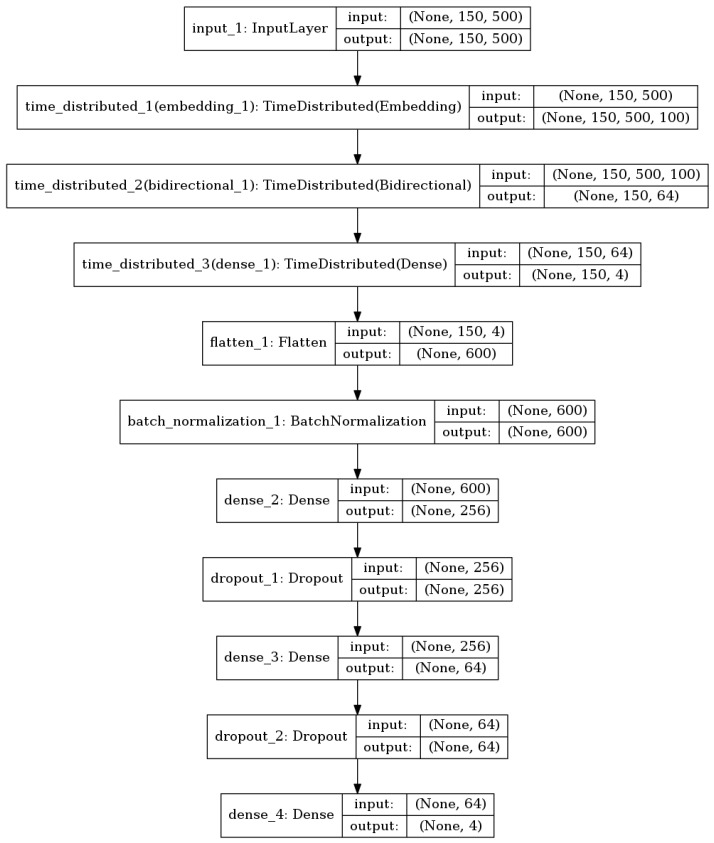
The network structure of the LSTM-based model.

**Figure 6 sensors-20-03645-f006:**
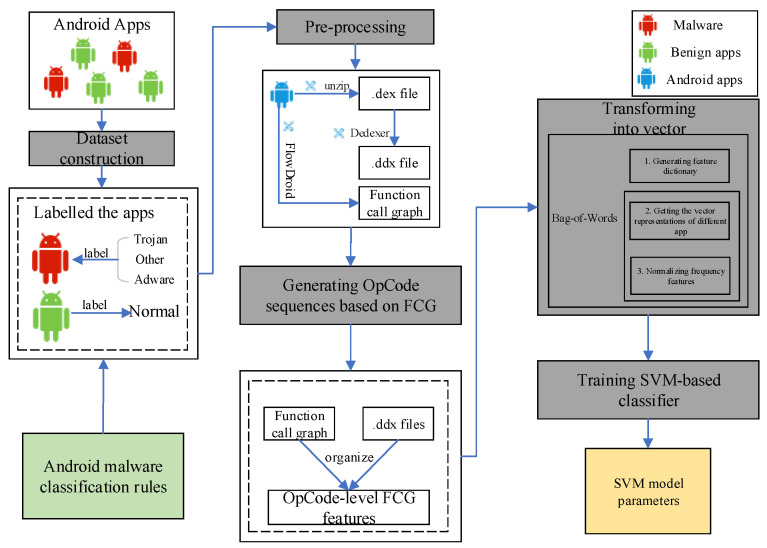
The training framework of the SVM-based classifier.

**Figure 7 sensors-20-03645-f007:**
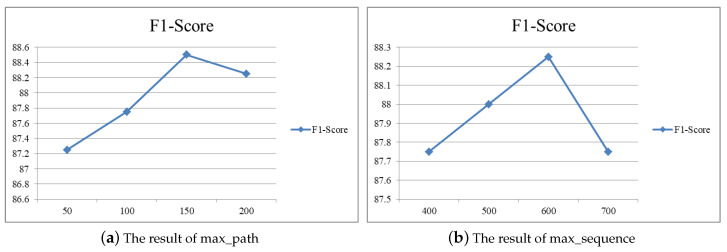
The result of different values of max_path and max_sequence. (**a**) The result of max_path; (**b**) The result of max_sequence.

**Figure 8 sensors-20-03645-f008:**
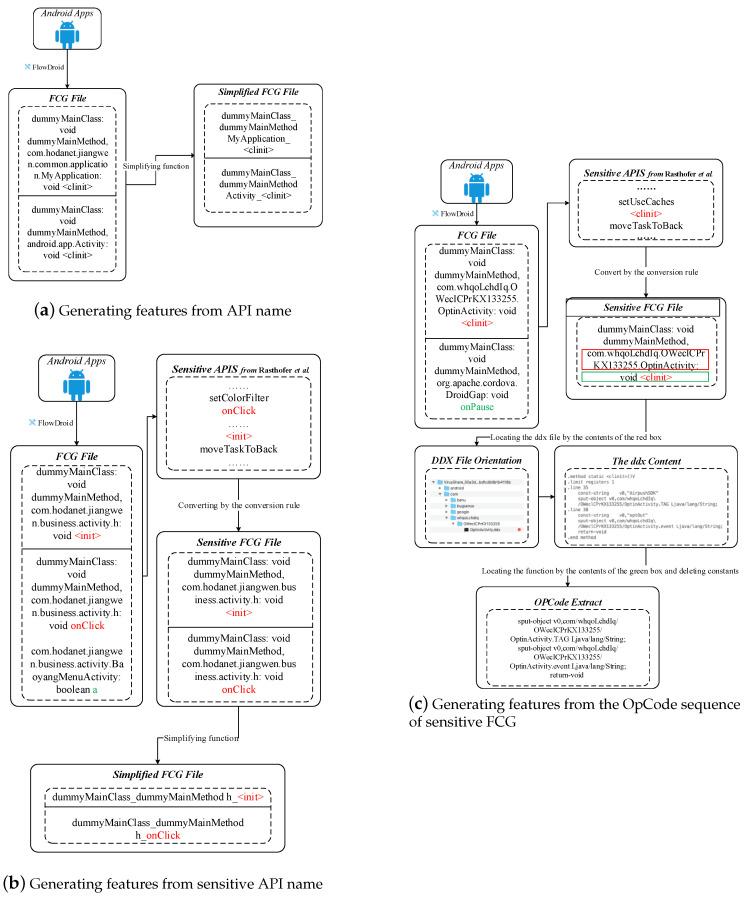
Illustrating the generating features from API name, sensitive API name, and OpCode sequence of sensitive FCG. (**a**) Generating features from API name; (**b**) Generating features from sensitive API name; (**c**) Generating features from the OpCode sequence of sensitive FCG.

**Figure 9 sensors-20-03645-f009:**
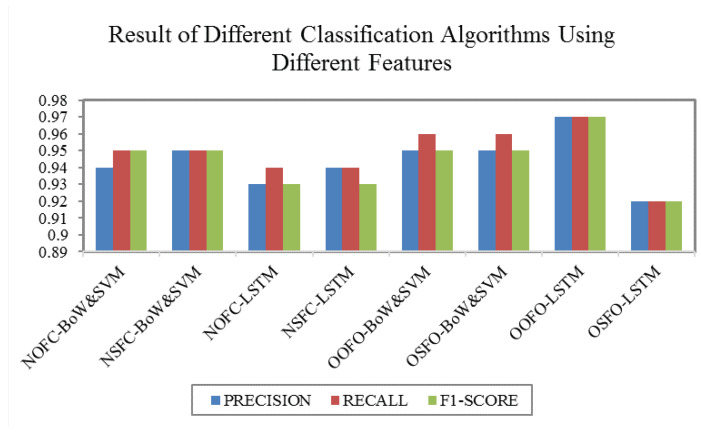
The result of different classification algorithms using different features.

**Figure 10 sensors-20-03645-f010:**
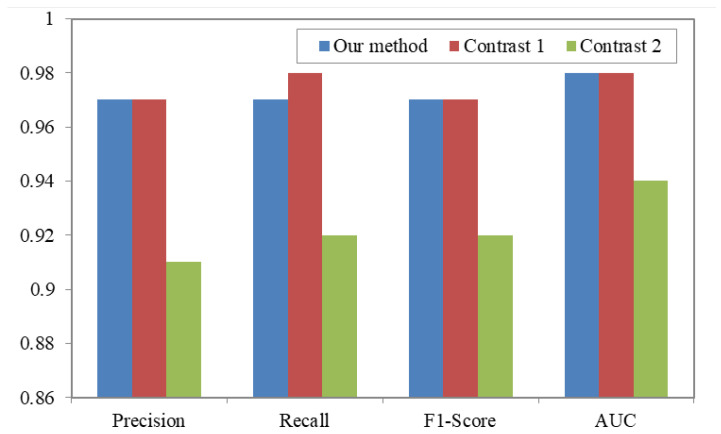
Comparative results of our proposed method and the two competing methods.

**Table 1 sensors-20-03645-t001:** Weights of prominent detection engines.

Engine	Weight
Alibaba	0.063
Tencent	0.06
Sophos AV	0.057
Antiy-AVL	0.054
McAfee	0.051
F-Secure	0.048
Avira	0.045
bitdefender	0.042
TrendMicro	0.039
GDsts	0.036
kaspersky	0.033
ikarus	0.03
AVG	0.027
Avast	0.024
AhnLab-V3	0.021
Qihoo-360	0.019
Kingsoft	0.016
AhnLab	0.013
McAfee-GW-Ec	0.01

**Table 2 sensors-20-03645-t002:** Experimental parameters for the LSTM-based classifier.

Parameter	Description	Value
unit	Dimensionality of the output space.	32
activation	Activation function	softmax
kernel_regularizer	Regularizer function applied to the kernel weights matrix	l2(0.01)
dropout	Fraction of the units to drop for the linear transformation of the inputs.	0.5
return_sequences	Whether to return the last output in the output sequence or the full sequence	FALSE
batch_size	The number of software to training fir each batch	16
epochs	The number of total training steps	100
max_nb_words	The number of valid words for embedding	20,000
max_path	The maximum number of extracted paths for each app	150
max_sequence	The maximum number of extracted words for each path	600

**Table 3 sensors-20-03645-t003:** Experimental parameters for the SVM-based classifier.

Parameter	Description	Value
C	Penalty parameter C of the error term.	1000
kernel	Specifies the kernel type to be used in the algorithm	linear
degree	Degree of the polynomial ker-nel function (‘poly’)	3
gamma	Kernel coefficient for ‘rbf’, ‘poly’ and ‘sigmoid’	auto
coef0	Independent term in kernel function. It is only significant in ‘poly’ and ‘sigmoid’	0
shrinking	Whether to use the shrinking heuristic	TRUE
probability	Whether to enable probability estimates	FALSE
tol	Tolerance for stopping criterion	0.001

**Table 4 sensors-20-03645-t004:** The result of different classification algorithms using different features.

Method	Precision	Recall	F1-Score
NOFC-BoW&SVM	0.94	0.95	0.95
NSFC-BoW&SVM	0.95	0.95	0.95
NOFC-LSTM	0.93	0.94	0.93
NSFC-LSTM	0.94	0.94	0.93
OOFO-BoW&SVM	0.95	0.96	0.95
OSFO-BoW&SVM	0.95	0.96	0.95
OOFO-LSTM	0.97	0.97	0.97
OSFO-LSTM	0.92	0.92	0.92

**Table 5 sensors-20-03645-t005:** The experimental results of Milosevic et al.’s method.

Class	Precision	Recall	bf F1-Score	bf AUC
0	0.95	0.99	0.96	0.98
1	0.98	0.98	0.98	0.99
2	0.98	0.96	0.97	0.98
Total	0.97	0.98	0.97	0.98

**Table 6 sensors-20-03645-t006:** The experimental results of Hou et al.’s method.

Class	Precision	Recall	F1-Score	AUC
0	0.89	0.93	0.91	0.94
1	0.91	0.91	0.91	0.94
2	0.94	0.92	0.93	0.95
Total	0.91	0.92	0.92	0.94

**Table 7 sensors-20-03645-t007:** Time consuming and model size of our proposed method and two comparative testing.

Method	Number of Samples	Total Time Consumption	Average Time Consumption	Model Size(M)
Our method	2796	11.3470	0.00406	6.17
comparative testing 1	2796	15.6182	0.00559	12.2
comparative testing 2	2796	1435.02	0.51324	566

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
