# Peer review of "OpCode-Level Function Call Graph Based Android Malware Classification Using Deep Learning"

_sensors, 2020, doi:10.3390/s20133645_

Round 1
Reviewer 1 Report
The authors present a novel methodology to detect Android Malware, using Function Call Graph (FCG) to capture the behaviour of the app and Deep Learning network to classify the app and analysing the opcodes of the app. They also provide a reliable methodology to build a dataset, that combines several tools to determine the more accurate label of each app.
The paper is well written and also is well structured. The authors provide schemas that help to get a more general view of the process. The methodology to evaluate and the presented results are also well detailed.
The primary issue with the presented methodology is that they classify the malware apps in two big categories: trojan and adware. They also added a new group of malware called "other". Although the authors justify their decision with the popularity of these prominent malware families, I would like to see how the systems perform with multiple categories (Deep Learning methods could do it well) or only with binary classification.
There are several minor issues that I think it could be significant to improve the paper:
- Why they use LSTM? There is an extensive and well-detailed explanation about how this network works, but not why they use them. I also miss a more detailed description of when this type of systems is used, why is the best configuration in this case.
- Some sentences need to be supported by references. For example, "Program language and natural language have some similarities, such as both have grammatical information and semantic information." This sound sentences should be backed with references.
- The authors provide the results using a bar-chart. It could be great if they also include the table (at least as an appendix, for example).
- It could be great if they include a brief explanation about what is one-hot encoding.
- Some references in the text should be reviewed (for example, Borja is the name of the author, not his surname).
Reviewer 2 Report
This paper proposes an approach to classify Android malwares based on the Operation Code-level Function Call Graph ((OpCode-FCG) as features and deep learning techniques, the LSTM model for classification. The authors collected a dataset of 2796 samples of normal and malware Android apps. The conducted experiments showed promising results that are comparable to two existing methods. The topic is of a general interest and the paper is easy to follow. However, it suffers from some issues that need to be addressed before publication.
The introduction is deprived of the recently deep learning-based methods related to the topic of study, consider adding the relevant work.
Please consider using a unified format to define abbreviations, first write the full words and then use abbreviations accordingly. Some abbreviations are mentioned without knowing what they refer for, such OpCode, RNN, NLP, etc.
When citing specific techniques such as RNN and LSTM, it is preferable to cite the original papers where these methods were proposed. And in case you point out their usage in the same topic of study, you can mention that and add the related work as in Ref. (22 and 23).
Also consider adding the references when the utilized tool or method mentioned for the first time as in AVTEST, VirusTotal, Euphony, etc.
Please, add more explanation on the experimental and evaluation protocols for dividing the dataset during training and reporting the performance metrics. This will help others to reproduce your results.
The references of some of the mentioned methods are missing. Also, consider revising the references and use the conferences names, and when citing a website add the access date.
Add suitable reference for shogun tool if possible.
Reviewer 3 Report
Nice paper. Clearly written, comprehensive in scope and adressing an important problem.
Also, the validation experiments are well described and provide convincing evidence as to the effectiveness of the OpCode proposed method.
I have two comments:
- The paper deals with malware and is aiming at indentifying new methods of attach. The paper does not refer to adverserial AI which is articular type of malware aiming at usign the system AI and affecting only its outcomes. The authors might want to expand their discussion to discuss such aspects.
- The experimental results are wide in scope but the authors should try to generalise them, beyond the scenarios they tested. In other words, some general claims on the performance of the propose method would be useful to have in the paper.
Reviewer 4 Report
1.- Language. The English could be improved, and the sections in the manuscript should be clearly explained.
2.- Introduction. The introduction is well structured and contains relevant information.
3.-References. References are adequately discussed in a fair context. Most of them are updated. Key references on the field are cited.
4.- Structure. The manuscript is well structured, containing all relevant sections required in a technical paper. The manuscript propose to use deep learning with static analysis of the OpCode-Level Function Call Graph to study complicated features of Android malware, where the obtained results are examined and explained correctly.
5.- Results and discussion. Results are appropriately discussed, compared, and presented in the appropriate context to findings from other recently published methods.
6.- Novelty/contribution. The manuscript presents fair novelty and contributions to the field:
There are some minor issues that authors need to addresses:
a) Provide the meaning of each used abbreviation just only in the first appearance through the text.
b) It is already known that Broad ANN has better overall performance than deep ANN in terms of its accuracy, bias, variance, convergence performance, etc. The authors could compare, for example, against a deep ANN method, and the advantages of their proposed method can be noticed.
c) Why do other malware samples such as Ransomware, Scareware, and SMS Malware are not classified by your proposed Android malware detection method?
d) How is the training error tolerance chosen? If the method is applied to a different malware, does the training error tolerance should be modified?
Round 2
Reviewer 2 Report
The authors addressed my comments.
Only a simple change is required related to the references:
Consider replacing the reference for RNNs to [1] instead of the wikipedia.org website, as that was the first paper to introduce the term recurrent nets.
[1] Rumelhart, David E; Hinton, Geoffrey E, and Williams, Ronald J (Sept. 1985). Learning internal representations by error propagation. Tech. rep. ICS 8504. San Diego, California: Institute for Cognitive Science, University of California.
